# Interpolymer Complexes of Eudragit^®^ Copolymers as Novel Carriers for Colon-Specific Drug Delivery

**DOI:** 10.3390/polym12071459

**Published:** 2020-06-30

**Authors:** Aleksandra V. Bukhovets, Nikoletta Fotaki, Vitaliy V. Khutoryanskiy, Rouslan I. Moustafine

**Affiliations:** 1Institute of Pharmacy, Kazan State Medical University, 16 Fatykh Amirkhan Street, 420012 Kazan, Russia; aleksandra.sitenkova@kazangmu.ru (A.V.B.); v.khutoryanskiy@reading.ac.uk (V.V.K.); 2Department of Pharmacy and Pharmacology, University of Bath, Claverton Down, Bath BA2 7AY, UK; n.fotaki@bath.ac.uk; 3Reading School of Pharmacy, University of Reading, Whiteknights, P.O. Box 224, Reading RG66AD, UK

**Keywords:** interpolymer complexes, Eudragit^®^, indomethacin, oral drug delivery, biorelevant media, controlled drug delivery, drug delivery systems

## Abstract

Interpolymer complexes (IPC) based on Eudragit^®^ EPO and Eudragit^®^ S100 were investigated as potential carriers for oral controlled drug delivery to the colon. IPC samples were prepared by mixing copolymer solutions in organic solvents (ethanol, isopropanol:acetone mixture (60:40, *% v*/*v*) and tetrahydrofuran). According to the data of elemental analysis, FTIR-spectroscopy, X-ray photoelectron spectroscopy and thermal analysis these IPCs have excess of anionic copolymer (Eudragit^®^ S100) in their structure; they are stabilized by hydrogen and ionic intermacromolecular bonds and do not include free copolymer domains. IPC have pH-independent swelling properties in the media mimicking gastrointestinal tract (GIT) conditions and provide colon-specific delivery of indomethacin in buffer solutions (pH 1.2; 5.8; 6.8; 7.4) and in biorelevant media (fasted state simulated gastric fluid, fasted state simulated intestinal fluid—version 2 and fasted stated simulated colonic fluid).

## 1. Introduction

Polymethacrylates of Eudragit^®^ series from Evonik Röhm GmbH are widely used for preparation of oral drug delivery systems [1,2,3,4]. Due to their structure Eudragit^®^-copolymers can participate in intermolecular interactions with other polymers and form interpolymer complexes (IPC). The driving forces of these interactions can be different and may result in formation of IPC with different properties [5,6,7,8]. There are several studies reporting the use of IPC based on Eudragit^®^ copolymers as drug carriers. Such carriers include IPC based on Eudragit^®^ copolymer with other chemically complementary macromolecules, e.g., Eudragit^®^ EPO and Carbopol^®^ [9,10,11,12,13], Eudragit^®^ EPO and sodium alginate [14,15,16], Eudragit^®^ (types RS, L, E) and Kollidon [17]. There are also IPC based on two oppositely charged Eudragit^®^ copolymers that can be used as drug delivery systems [1,18,19,20,21,22,23].

In the last decades, many researchers have focused their attention on colon specific drug delivery systems [24,25,26,27,28,29,30,31]. These systems have been developed for local treatment of different inflammatory bowel diseases, such as Crohn’s disease, ulcerative colitis, colon cancer, local treatment of colonic diseases and systemic delivery of protein and peptides [32]. There are several strategies for achieving colon-specific targeting including 3D-printed systems [33], enzyme-responsive systems [34,35], microparticulate [36,37,38] and nanoparticulate systems [39,40], film- and press-coated formulations [41,42,43,44]. Eudragit^®^ copolymers also can be used in colon-specific drug delivery systems. Eudragit^®^ S dissolves at pH > 7 and is used for colon targeting. The colon-specific drug delivery systems using Eudragits^®^ have been described for different drugs such as 5-aminosalicylic acid, insulin, prednisolone, etc. [4,45,46,47]. However, some reports revealed that the use of Eudragit^®^ S100 alone is not suitable for colon targeted drug delivery due to transit variability [32]. In order to solve this problem, it is possible to use interpolymer complexes based on complementary Eudragit^®^ copolymers [19].

The aim of this study was to prepare IPC based on Eudragit^®^ EPO and Eudragit^®^ S100 copolymers as carriers for colon-specific delivery of indomethacin—a non-steroidal anti-inflammatory drug that has chemoprotective effects against tumors, reducing the risk of colon cancer [48,49,50]. Unlike most previous studies reporting the formation of IPC between Eudragit^®^ EPO and Eudragit^®^ S100 in aqueous media, the present work was focused on the use of organic solvents (isopropanol:acetone mixture, ethanol and tetrahydrofuran). This approach has led to the IPC, whose compositions and properties are substantially different from the polycomplexes formed from these copolymers in aqueous media. One of these particularly unique and valuable properties is the stability of these IPC in the media with different pHs representing different parts of gastrointestinal tract.

## 2. Materials and Methods

### 2.1. Materials

Eudragit^®^ EPO (EPO) is a cationic terpolymer of *N*,*N*-dimethylaminoethyl methacrylate with methylmethacrylate and butylmethacrylate (molar ratio 2:1:1, MW~150 kDa). Eudragit^®^ S100 (S100) is an anionic copolymer of methacrylic acid with methylmethacrylate (mole ratio 1:2, MW~135 kDa). Different types of Eudragit^®^ (EPO, S100) were generously donated by Evonik Röhm GmbH (Darmstadt, Germany). These copolymers were used after vacuum drying at 40 °C for 2 days. Indomethacin (IND) purchased from Sigma (Bornem, Belgium) was used as a model drug. Potassium dihydrogen phosphate, potassium hydrogen diphosphate, hydrochloric acid, sodium chloride and sodium hydroxide were purchased from Sigma-Aldrich (Irvine, UK). Pepsin from porcine gastric mucosa, maleic acid and sodium taurocholate were purchased from Sigma Chemical (St Louis, USA). Egg phosphatidylcholine was from Lipoid GmbH (Ludwigshafen, Germany). Bovine serum albumin was from Fluka Chemie GmbH (Munich, Germany). All other chemicals were of analytical grade, except for solvents which were of HPLC grade.

### 2.2. Preparation of Solid IPCs

EPO and S100 solutions were prepared by dissolving the copolymers in ethanol, tetrahydrofuran and isopropanol–acetone mixture (60:40, *% v*/*v*). These solutions were mixed at a constant temperature (25 °C) at different molar ratios (Table 1). Formation of the IPC occurred immediately upon mixing these solutions, which initially resulted in cloudy colloidal solution, followed by aggregation of primary polycomplex particles and their eventual precipitation. The IPC samples were prepared in a laboratory reactor system LR 1000 control under continuous and simultaneous agitation at 1000 rpm during 2 h using Eurostar 60 control overhead stirrer (IKA^®^ Werke GmbH, Staufen, Germany). After isolation of the precipitates of IPC particles from solutions by centrifugation (10 min, 3000 rpm, centrifuge ELMI, Latvia), they were washed three times with ultrapure water (Smart2Pure UV/UF, Thermo Scientific, Waltham, MA, USA.), and subsequently vacuum-dried (vacuum oven VD 23, Binder, Germany) for 2 days at 40 °C until constant weight. The solid samples were ground with a grinder, ball milled (IKA® Ultra Turrax^®^ Tube Drive P control Workstation, 10 min, 2000 rpm) and stored in tightly sealed containers at room temperature.

### 2.3. Elemental Analysis

The compositions of the dried IPC samples were investigated by elemental analysis using a CHNS/O Elemental analyzer Thermo Flash 2000 (Thermo Fisher Scientific, Paisley, UK) and calculated as Z = [EPO]:[S100] (mol/mol). The vacuum dried samples (at 40 °C for 2 days) were weighed into a crucible on a XP6 Excellence Plus XP micro balance (Mettler Toledo, Greifensee, Switzerland). The crucibles with samples were packed and placed into the combustion reactor via autosampler. Temperature in the oven was 900 °C, and a gas flow rate was 10 mL/min. Calibration of the instrument was performed with atropine standard (Thermo Fisher Scientific, Paisley, UK). Eager Xperience Data Handling Software was used to analyze the results.

### 2.4. Fourier Transformed Infrared (ATR-FTIR) Spectroscopy

ATR-FTIR spectra were recorded by a Nicolet iS5 FTIR spectrometer (Thermo Scientific, Waltham, MA, USA) using the iD5 smart single bounce ZnSe ATR crystal. The spectra were analyzed using OMNIC spectra software.

### 2.5. X-ray Photoelectron Spectroscopy (XPS)

XPS measurements were carried out using a spectrometer K-Alpha (Thermo Fisher Scientific, Paisley, UK). The surface of the samples was recharged during this analysis as a result of their low electrical conductivity. Registration of spectra was carried out using a combined ion–electron compensation gun.

### 2.6. Thermal Analysis

Modulated DSC (mDSC) measurements were carried out using a Discovery DSC™ (TA Instruments, New Castle, DE, USA), equipped with a refrigerated cooling system (RCS90). TRIOS™ software (version 3.1.5.3696) was used to analyze the DSC data (TA Instruments, New Castle, DE, USA). Tzero aluminum pans (TA Instruments, New Castle, DE, USA) were used in all calorimetric studies. The empty pan was used as a reference and the mass of the reference pan and of the sample pans were taken into account. Dry nitrogen was used as a purge gas through the DSC cell at 50 mL/min. Indium and n-octadecane standards were used to calibrate the DSC temperature scale; enthalpic response was calibrated with indium. Calibration of heat capacity was done using sapphire. Initially the samples were cooled from room temperature to 0 °C, then kept at 0 °C for 5 min and analyzed from 0 to 250 °C. The modulation parameters used were: 2 °C/min heating rate, 40 s period and 1 °C amplitude. Glass transition temperatures were determined using the reversing heat flow signals. All measurements were performed in triplicate.

### 2.7. Preparation of IPC Compacts

In order to determine the degree of swelling, flat-faced 100 mg IPC compacts with 8 mm diameter were prepared by compressing the given amount of powder at 2.45 MPa using a hydraulic press (PerkinElmer^®^, Waltham, MA, USA).

For dissolution testing, flat-faced 150 mg compacts (100 mg of IND and 50 mg IPC mixture) and 8 mm diameter were prepared by powder compression at 2.45 MPa using a hydraulic press (PerkinElmer^®^, Waltham, MA, USA).

### 2.8. Swelling Studies

Swelling was investigated under conditions, mimicking the gastrointestinal tract up to the colon: the first hour in simulated gastric medium (0.1-M HCl; pH 1.2), then the pH of the medium was gradually increased using phosphate buffers: pH 5.8 for the next two hours, pH 6.8 for a further two hours, and finally pH 7.4 was maintained until the end of the experiment (a further two hours) [51].

The polymeric compact was placed in a tarred basket, which was immersed into a thermostatic bath (37.0 ± 0.5 °C). The total volume of the medium was 40 mL. The basket was removed from the medium every 15 min and the compact was carefully dried using a filter study and weighed.

The degree of swelling (H_%_) was calculated using the following equation:H_%_ = ((m_2_ − m_1_)/m_1_)∙100(1)
where m_1_ is the weight of the dry sample and m_2_ is the weight of the swollen sample.

### 2.9. Preparation of Biorelevant Media

Fasted state simulated gastric fluid (FaSSGF) and fasted state simulated intestinal fluid (FaSSIF-V2) were prepared according to Jantratid et al. [52]. Fasted State Simulated Colonic Fluid (FaSSCoF) was prepared according to Vertzoni et al. [53]. The composition of the biorelevant media is presented in Appendix A.

### 2.10. Indomethacin Release Studies

The release of IND was investigated in buffer solutions (pH 1.2; 5.8; 6.8 and 7.4) and in biorelevant media (FaSSGF, FaSSIF-V2 and FaSSCoF) for 7 h with sequential media change using the BIO-DIS reciprocating cylinder apparatus (USP Apparatus III; Agilent Technologies, Santa Clara, CA, USA) in 200 mL of medium and the Flow through cell apparatus (USP Apparatus IV; CE 7 smart USP Apparatus IV; SOTAX, Aesch, Basel, Switzerland) in 500 mL of medium. The conditions used in the dissolution studies in simple buffers and in biorelevant media are shown in Appendix A, respectively.

The concentration of IND in the samples from the dissolution tests was determined with UV-spectrophotometry (Helios, Thermo Electron Corporation, Waltham, MA, USA) in the buffer solutions samples at 262.5 nm (pH 1.2) and 265 nm (pH 5.8; 6.8; 7.4) and with HPLC-UV (Agilent 1200 Series, Agilent Technologies, Santa Clara, CA, USA) in the biorelevant media samples. The concentration of IND in the release medium was calculated based on calibration curves (Appendix A). Release profiles were modeled mathematically using Origin^®^ (scientific graphing & analysis software, Version 7.5, Origin Lab Corp., Northampton, MA, USA).

The release data were fitted according to Korsmeyer–Peppas equation which combines Fickian diffusion and Case-II transport [54]:M_t_/M_∞_ = k⋅t^n^(2)
where M_t_ is the amount of drug released at time t, M_∞_ is the total amount of drug, k is the apparent release rate constant, which includes structural and geometric characteristics of the compact, n is the exponent of release, showing the drug transport mechanism.

## 3. Results

### 3.1. Evaluation of Structure and Composition of IPC

Interactions between oppositely charged polymers are traditionally studied in aqueous media, which favors electrostatic attraction forces and also additional stabilization of interpolyelectrolyte complexes by hydrophobic effects resulting from the presence of nonpolar groups in both polymers [6,55]. The interactions between weak polyacids and non-ionic proton-accepting polymers in aqueous solutions typically result in hydrogen-bonded complexes, which are additionally stabilized by hydrophobic effects [6,55]. However, when the solvent medium is switched from aqueous to a fully or partially organic then this may substantially affect the intensity of interactions, the structure of polycomplexes formed and in some cases, it may even prevent the formation of IPC [56,57].

EPO and S100 are amphiphilic copolymers that have substantial quantity of hydrophobic groups in their structure. This makes the solubility of these copolymers in water quite limited by specific pH windows and requires some adjustments to achieve interpolymer complexation in aqueous media [19]. However, there is also a possibility of forming these IPCs in fully organic media. The first use of organic medium (isopropanol/acetone 60:40) to form polycomplexes from EPO and S100 was reported by Gallardo et al. [1,18]; however they mostly focused their research on the pharmaceutical aspects of these materials and did not study the changes in the nature of the complexation observed upon the switch from aqueous to organic media. In the present study, we have studied the complex formation between EPO and S100 in three different organic solvents (isopropanol/acetone 60:40 mixture, pure ethanol as well as tetrahydrofuran) and evaluated their effect on the complexation. These solvents were selected because they could be used to dissolve both EPO and S100; they are also volatile and can be easily removed by vacuum-drying.

The composition and the thermal properties (glass transition temperatures (*Tg*)) of the IPC formed in organic solvents after their precipitation and isolation were determined using elemental analysis. Additionally, we have studied the effect of the order of polymers mixing. The results of these studies are summarized in Table 1.

It can be clearly seen that all IPC samples have excess of anionic copolymer (S100) in their structure. This is quite different from the compositions of the IPC formed from EPO and S100 in aqueous solutions, which was reported in our previous publication [19]. The IPC formed from these copolymers using similar ratios of reagents in aqueous solutions have a slight excess of EPO in their structure ([EPO]:[S100] = 1.26:1). This is clearly indicating that the switch of solvents from aqueous to organic ones has a significant effect on the composition and possibly the structure of the IPC.

A comparison between the compositions of the IPC prepared from organic solvents in the present work indicates that this is practically not affected by the order of copolymer addition. For example, the addition of S100 to EPO at 1:1.5 ratio results in formation of a polycomplex with 1:1.94 copolymer ratio, whereas the reverse addition of EPO to S100 gives 1:2.13. Additionally, our data indicate that the excess of S100 in the starting copolymer ratio leads to a greater amount of S100 incorporated into the IPC. Changes in the solvent nature also do not show any substantial effects on the compositions of the IPC.

The thermal properties of the IPC studied by DSC clearly show the presence of a single Tg point, which indicates that the copolymers are miscible with each other at a molecular level, confirming the fact that IPC is a new individual compound, whose properties are entirely different from the properties of the parent copolymers.

FTIR-spectroscopy was used for estimation of the interactions between two copolymers. According to our previous results, the FTIR-spectra of pure Eudragit^®^ EPO show the presence of two characteristic bands at 2770 and 2820 cm^−1^ corresponding to valence vibrations of non-ionized dimethylamino groups. Additionally, there are two bands at 1705 and 1730 cm^−1^ corresponding to carboxylic and ester groups, respectively, in the case of Eudragit^®^ S100 copolymer [19].

FTIR-spectra of IPC samples are characterized by the presence of all bands mentioned above, but intensity of bands at 2770 and 2820 cm^−1^ is decreased (Figure 1).

Decrease can be related to interpolymer interactions between dimethylamino groups of Eudragit^®^ EPO and carboxylic groups of Eudragit^®^ S100. In addition, the band at 1560 cm^−1^ is appeared in the spectra of IPCs. According to the literature, this band is related to intermacromolecular ionic bonds in polycomplexes [19,22]. However, taking into account that the organic solvents were used in the preparation of these IPC the presence of intermacromolecular hydrogen bonds between the copolymers can also be suggested. Additional confirmation for the presence of hydrogen bonding is seen from a wide band at 3270 cm^−1^.

XPS was used additionally to confirm the nature of bonding in all IPC samples. The N1s spectra of the IPC samples are characterized by several peaks (Figure 2), indicating that the nitrogen-containing groups are present in several forms: the ground state (399.5 eV) is neutral amino groups; also, there are groups with hydrogen bonds (400.5 eV, chemical shift from neutral groups +1.0 eV) and groups in ionic form (401.7 eV, chemical shift from neutral groups +2.2 eV). These results further confirm that the interpolymer complexes are stabilized by both hydrogen bonds and ionic intermacromolecular bonds.

### 3.2. Thermal Analysis

Thermal analysis such as differential scanning calorimetry is an excellent tool to study interpolymer complexes as it may provide the confirmation that the product of interaction between the polymers is a new individual compound rather than a physical mixture of two individual copolymers [19]. Figure 3 shows the DSC thermograms of individual copolymers and IPCs. Tg for EPO and S100 are observed at 54.7 and 173.8 °C, respectively. These values are in good agreement with the values reported in the literature [19].

IPC samples are characterized by the presence of a single Tg within the range of Tg typical for individual copolymers. This observation proves the complete molecular miscibility between the copolymers within the interpolymer complex. The value of Tg is linearly increased with an increase in the amount of Eudragit^®^ S100 in the IPC structure (Appendix A). This result was expected as Eudragit^®^ S100 is a more rigid polymer and it has greater Tg value (173.8 ± 1.6 °C) compared to Eudragit^®^ EPO (Tg = 54.7 ± 1.3 °C). The presence of greater quantity of more rigid polymer in the IPC results in a greater glass transition temperature exhibited by these samples. This is in agreement with our previous observations [19,22]. The summary of all Tg values for different IPC samples is presented in Table 1.

### 3.3. Swelling Properties

In order to study the behavior of pharmaceutical dosage forms based on these IPCs we have prepared drug-free compacts and studied their swelling behavior in the media mimicking different parts of gastrointestinal tract. The preliminary experiments indicated that the compacts based on IPC-1, IPC-2 and IPC-3 have demonstrated the greatest stability in the dissolution media therefore all subsequent detailed swelling studies were conducted specifically with these samples.

It is known that Eudragit^®^ EPO is soluble and Eudragit^®^ S100 is dispersible in the acidic medium [19]. Evaluation of swelling properties of IPC-1 and IPC-2 samples indicates that they preserve their shape and show only very minor swelling in the media simulating gastrointestinal tract conditions. However, the compact based on IPC-3 is undergoing more substantial changes during this experiment due to the swelling followed by partial surface erosion (Figure 4).

Figure 5 shows the swelling profiles of IPC compacts in buffer solutions mimicking the pH values in different parts of gastrointestinal tract. These profiles were recorded using gravimetric technique by measuring weight of compacts at different swelling times. It can be seen that the swelling behavior of IPC-1 and IPC-2 is very close to each other and can be described as pH-independent. IPC-3 has similar swelling profile, but only in the first two media (pH 1.2–1 h and pH 5.8–2 h) with an increase in the swelling index (approximately up to 800%) in the medium with pH 6.8. Then, after moving to the last medium (pH 7.4), a two-times decrease in the degree of swelling (to approximately 400%) is observed due to the visible surface erosion (Figure 4). This behavior of IPC-3 is possible related to its composition, which contains the larger quantity of EPO compared to IPC-1 and IPC-2.

Monitoring the compositional and structural changes in the IPC compacts was carried out by the methods of elemental analysis and FTIR-spectroscopy, respectively. These methods were successfully used previously for Eudragit^®^ EPO/Eudragit^®^ L100 [22], Eudragit^®^ EPO/Eudragit^®^ L100-55 and other polycomplex systems [20]. According to the results generated there are no significant changes in the composition of interpolymer complexes in buffer solutions mimicking the pH values of different parts of gastrointestinal tract (Table 2 and Appendix A).

The composition of IPC-3 changed from [EPO]:[S100] (mol/mol) 1:2.13 at the start of the experiment to 1:2.30 in the buffer solution with pH 7.4 with the loss of EPO amount due to the surface erosion, that was of course predictable due to the results described previously [22].

Figure 6 shows the FTIR-spectra of interpolymer complex compacts (for IPC-1) after swelling in buffer solutions mimicking the pH values in different parts of gastrointestinal tract.

According to the spectral data the contribution of intermolecular ionic bonds was reduced in acidic medium and then these bonds were restored again when the pH value reached 7.4 (for IPC-2 and IPC-3 the spectra were similar; data not shown); this was in good agreement with the results reported in our previous studies [19,20,22]. The lack of changes in the compositions of IPC during the transit of compacts through the media of different pHs indicated that macromolecules were tightly bound to each other and this could be related to combined effects from intermacromolecular ionic attraction and hydrogen bonding. This behavior of IPC prepared from organic solvents was substantially different from the samples that were prepared from aqueous salt-containing media reported in our previous studies [19,20,22]. It was likely that this difference was related to additional stabilization of IPC formed in organic solvents by intermacromolecular hydrogen bonding. Ethanol, isopropanol and tetrahydrofuran were organic solvents, where formation of hydrogen-bonded IPC was reported previously for a variety of polymeric pairs [58,59,60,61]. The stability of IPC formed by EPO and S100 in organic solvents at different pHs may open up an interesting opportunity for their pharmaceutical application in colon-specific drug delivery. These IPC may provide protection for the active ingredient during the dosage form transit.

### 3.4. Release of Indomethacin from IPC Compacts

Assessment of release of a model drug IND was carried out using USP Apparatus III and IV to evaluate the potential of application of interpolymer complexes as carriers for controlled drug delivery to the colon. The release of IND from the interpolymer complexes with the USP Apparatus IV (flow through cell apparatus) was negligible in the buffer media with pH value lower than 6.8 and in FaSSGF (Figure 7 and Figure 8).

IND was released only in the last two media with pH values of 6.8 and 7.4 in the case of the study of IND in buffer solutions and in FaSSIF-V2 and FaSSCoF in the case of the study of IND in biorelevant media. This release profile reveals that these formulations have gastro-resistant properties and release the drug mostly in the lower part of the intestine. It is important to note that the release of IND from the formulations based on IPC-2 and IPC-3 was similar despite their different compositions and the conditions used for their preparation. In the case of the release evaluation with the Bio Dis apparatus (USP III) the release profiles of all complexes exhibited their gastro-resistant properties and released IND in the lower part of the intestine media only (Figure 9 and Figure 10).

The release of IND from the formulations of IPC-2 and IPC-3 was also similar in the studies with the BioDis apparatus, as observed in the studies with the flow-through cell apparatus. A higher release of IND was observed with the BioDis apparatus in all conditions tested due to the disintegration of the compacts. Under all conditions tested, release of IND from formulations based on IPC-1 was higher compared to IND release from IPC-2 and IPC-3 formulations.

The results of the release studies with the flow-through cell apparatus and with the Bio Dis apparatus with sequential media change confirms that all the IPCs tested are erosion-type systems.

Table 3 and Table 4 summarize the data on the release exponent (n) and the proposed transport mechanism applicable to the studied samples using USP III apparatus in simple buffer solutions and in biorelevant media, respectively.

According to these results, the release of IND from all compacts corresponded to Super Case-II transport mechanism which characterizes non-swelling eroding systems. In this case, drug release occurred due to surface erosion of the compact, which confirmed our assumption about the mechanism of IND transport from the IPC, and also explained the fact that in both buffer and biorelevant media, the rate of drug release process, tested in USP apparatus III was higher compared to USP apparatus IV.

It is known that drug solubility in the biorelevant media is increased compared to the solubility determined in aqueous buffer solution, as a result of enhanced wetting and/or micellar solubilization of poorly soluble drugs [62]. Yazdanian et al. investigated non-steroidal anti-inflammatory drugs (indomethacin, sulindac, ibuprofen and naproxen) and showed an increased drug solubility in FaSSIF compared to buffer solution [63]. However, in our case there were similar mechanisms of drug release in buffer solutions and biorelevant media that could be explained by the influence of IPC on release behavior of IND. This confirms that all IPC could potentially be used as carriers for colon-specific drug delivery systems.

## 4. Conclusions

According to FTIR-spectroscopy and XPS results the interpolymer complexes based on Eudragit^®^ EPO and Eudragit^®^ S100 prepared in ethanol, isopropanol/acetone and tetrahydrofuran mixtures were stabilized by cooperative system of hydrogen and ionic intermacromolecular bonds. All samples of IPC had an excess of anionic copolymer—Eudragit^®^ S100 in their structure. The copolymers were fully miscible within the IPC.

The compacts prepared from IPC-1 and IPC-2 had pH-independent swelling properties in the media mimicking GIT conditions. IPC-3, on the contrary, exhibits pH-dependent swelling properties. All tested IPC samples provide colon-specific delivery of indomethacin in buffer solutions (pH 1.2, 5.8, 6.8, 7.4) as well as in biorelevant media (FaSSGF, FaSSIF-V2, FaSSCoF) mimicking GIT conditions.

The use of organic solvents for preparing IPC based on Eudragits^®^ not only leads to the new pharmaceutical materials with unique physicochemical properties, but also made their application more technologically relevant. Dissolution of Eudragits^®^ in organic solvents was a straightforward process that did not require any further adjustments and resulted in solutions with greater concentrations. The preparation of aqueous solutions of Eudragits^®^ was more complicated as it required adjustment of pH.

Future research may focus on the evaluation of stability of these formulations and the effects of storage conditions on the drug release profiles. Further studies could also be focused on the particle size effects on the drug release profiles and investigation of other physicochemical properties of IPC compacts.

## Figures and Tables

**Figure 1 polymers-12-01459-f001:**
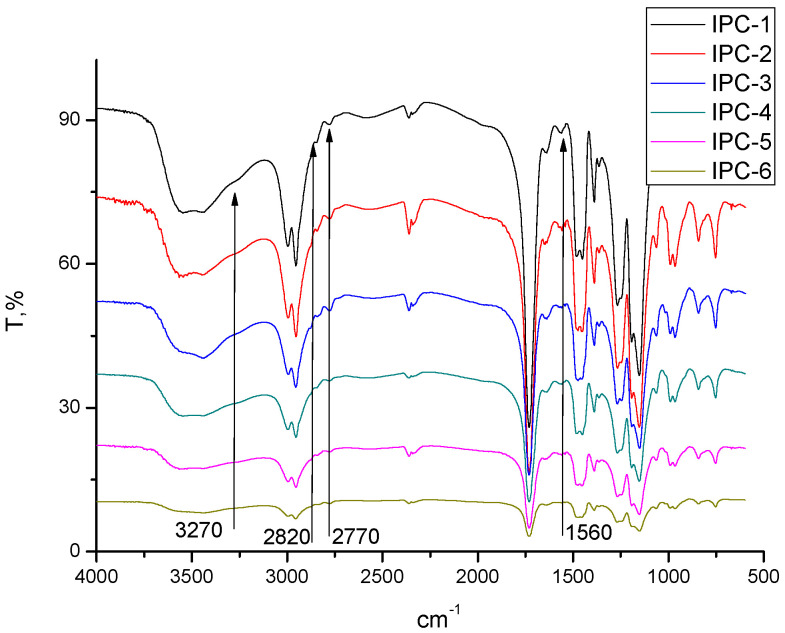
FTIR spectra of the IPC.

**Figure 2 polymers-12-01459-f002:**
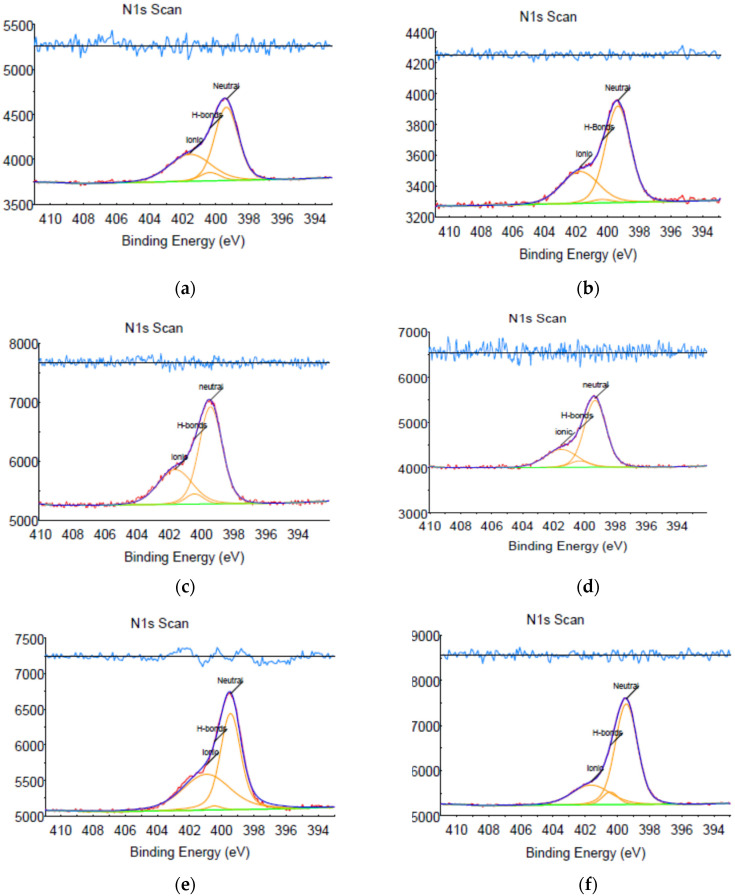
N1s spectra of interpolymer complexes. (**a**) IPC-1; (**b**) IPC-2; (**c**) IPC-3; (**d**) IPC-4; (**e**) IPC-5; (**f**) IPC-6.

**Figure 3 polymers-12-01459-f003:**
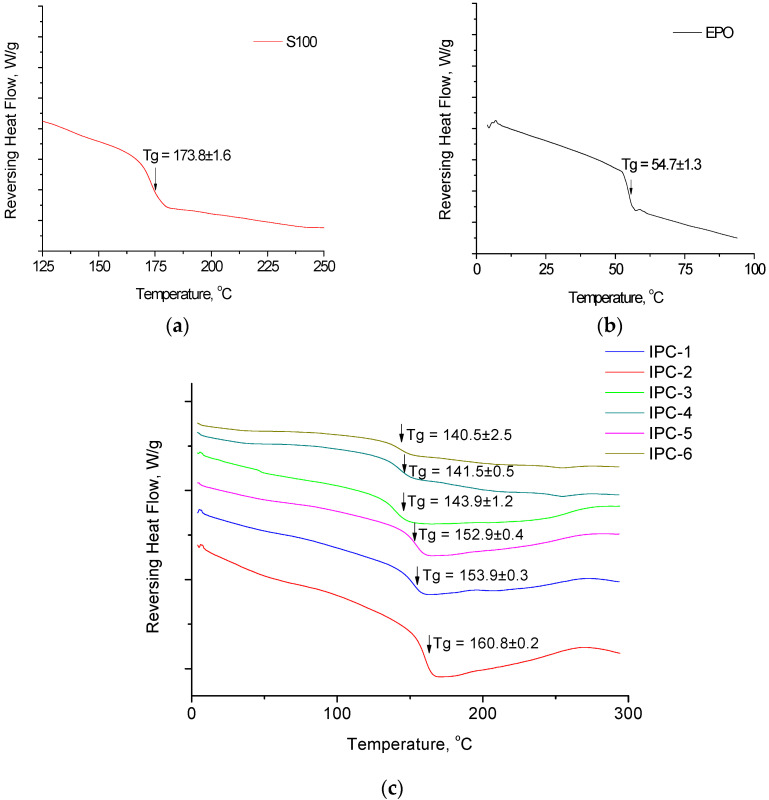
Exemplar differential scanning calorimetry (DSC) thermograms of interpolymer complexes and Eudragit^®^ copolymers: (**a**) Eudragit^®^ S100; (**b**) Eudragit^®^ EPO and (**c**) IPCs.

**Figure 4 polymers-12-01459-f004:**
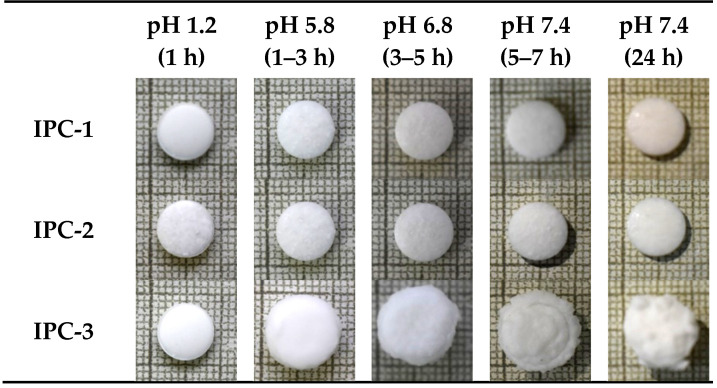
External appearance of IPC compacts during the swelling test.

**Figure 5 polymers-12-01459-f005:**
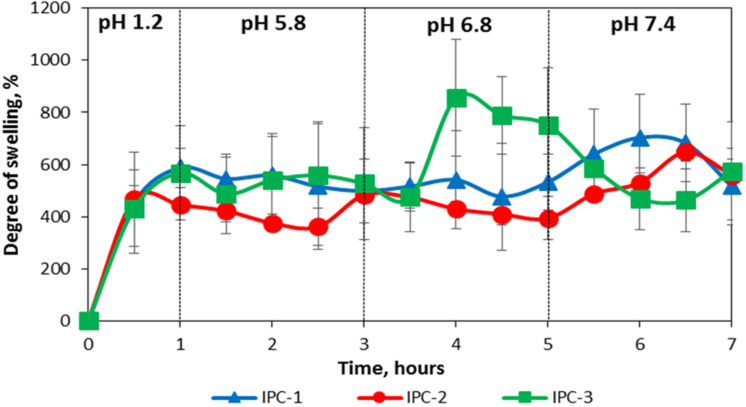
Swelling profiles of IPC compacts in buffer solutions mimicking the pH values in different parts of gastrointestinal tract.

**Figure 6 polymers-12-01459-f006:**
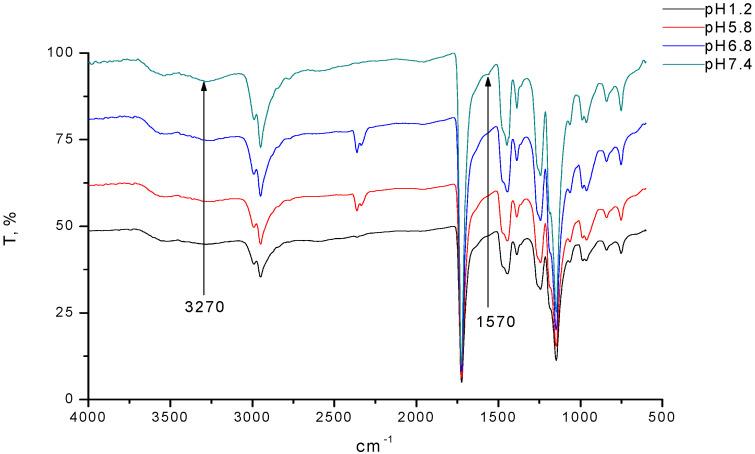
FTIR-spectra of polycomplex compacts (prepared from IPC-1), after swelling in buffer solutions mimicking the pH values in different parts of gastrointestinal tract.

**Figure 7 polymers-12-01459-f007:**
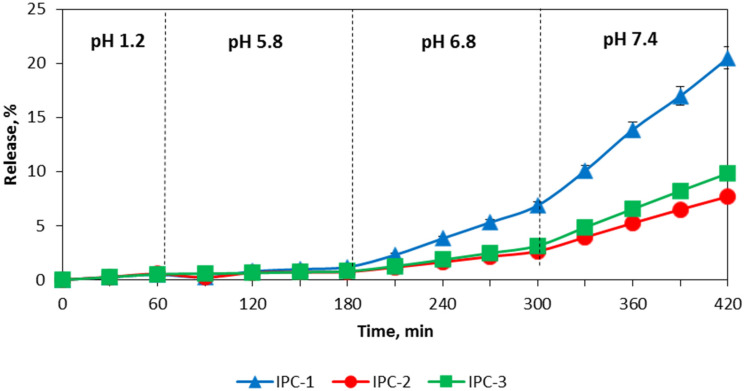
Release profiles of indomethacin from IPC compacts in the buffer solutions mimicking the environment of gastrointestinal tract (USP IV apparatus).

**Figure 8 polymers-12-01459-f008:**
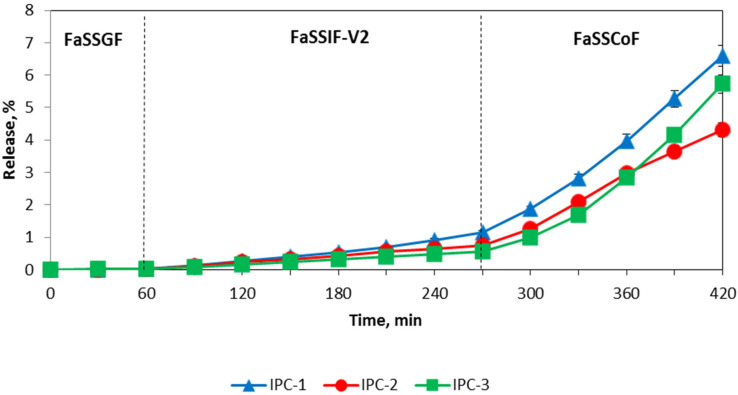
Release profiles of indomethacin from IPC compacts in the biorelevant media (USP IV apparatus).

**Figure 9 polymers-12-01459-f009:**
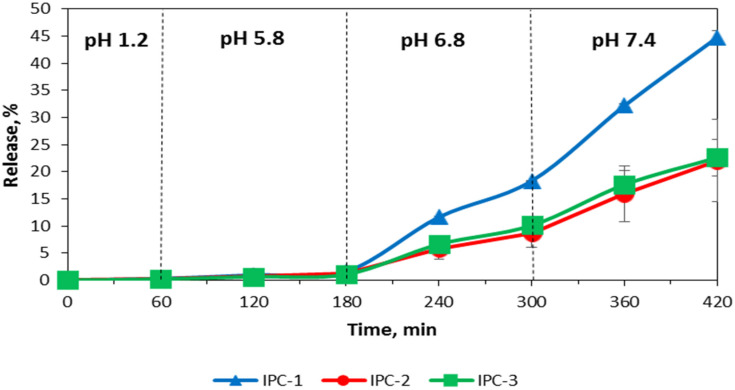
Release profiles of indomethacin from IPC compacts in the buffer solutions mimicking the environment of gastrointestinal tract (USP III apparatus).

**Figure 10 polymers-12-01459-f010:**
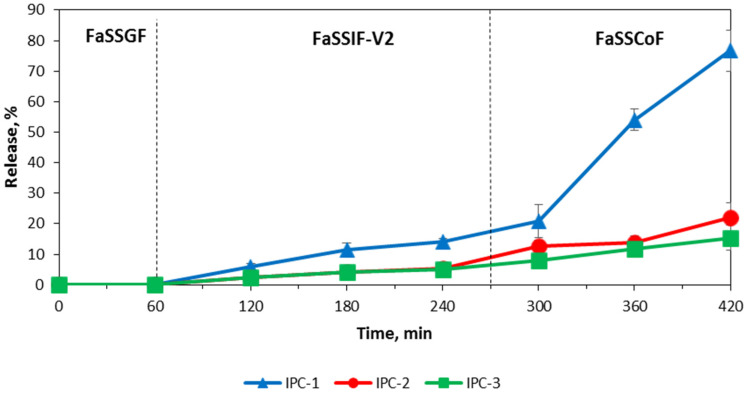
Release profiles of indomethacin from IPC compacts in the biorelevant media (USP III apparatus).

**Table 1 polymers-12-01459-t001:** Characteristics of interpolymer complexes (IPC) samples.

IPC	Solvent	Polymer Ratio in Mixture [EPO]:[S100], (mol/mol)	Order of Mixing	Composition of IPC [EPO]:[S100], (mol/mol)	*Tg*, °C
IPC-1	isopropanol/acetone	1:2	EPO to S100	1:2.96	153.9 ± 0.3
IPC-2	isopropanol/acetone	1:3	S100 to EPO	1:3.68	160.8 ± 0.2
IPC-3	ethanol	1:1.5	EPO to S100	1:2.13	143.9 ± 1.2
IPC-4	ethanol	1:1.5	S100 to EPO	1:1.94	141.5 ± 0.5
IPC-5	tetrahydrofuran	1:2	EPO to S100	1:2.58	152.9 ± 0.4
IPC-6	tetrahydrofuran	1:1.5	S100 to EPO	1:1.90	140.5 ± 2.5

**Table 2 polymers-12-01459-t002:** Results of elemental analysis of IPC compacts for N content after their swelling in buffer solutions mimicking the pH values in different parts of gastrointestinal tract.

Sample	pH	N%	Composition of IPC [EPO]:[S100], (mol/mol)
IPC-1	1.2	1.60 ± 0.01	1:3.15
	5.8	1.64 ± 0.01	1:3.09
	6.8	1.65 ± 0.02	1:3.08
	7.4	1.68 ± 0.02	1:2.99
IPC-2	1.2	1.44 ± 0.00	1:3.73
	5.8	1.49 ± 0.02	1:3.57
	6.8	1.59 ± 0.01	1:3.24
	7.4	1.52 ± 0.02	1:3.44
IPC-3	1.2	1.92 ± 0.17	1:2.44
	5.8	1.90 ± 0.01	1:2.46
	6.8	1.93 ± 0.05	1:2.41
	7.4	1.99 ± 0.01	1:2.30

**Table 3 polymers-12-01459-t003:** Mathematical modeling of indomethacin release from IPC compacts in buffer solutions using USP III apparatus.

	Compact Characteristics		
	IPC-1	IPC-2	IPC-3
Release exponent (*n*)	2.67 ± 0.20	2.65 ± 0.18	2.45 ± 0.24
*R* ^2^	0.9906	0.9927	0.9846
Transport mechanism	Super Case-II	Super Case-II	Super Case-II

**Table 4 polymers-12-01459-t004:** Mathematical modeling of indomethacin release from IPC compacts in biorelevant media using USP III apparatus.

	Compact Characteristics		
	IPC-1	IPC-2	IPC-3
Release exponent (*n*)	2.98 ± 0.37	2.08 ± 0.25	1.76 ± 0.12
*R* ^2^	0.9736	0.9737	0.9910
Transport mechanism	Super Case-II	Super Case-II	Super Case-II

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
