# Peer review of "Interpolymer Complexes of Eudragit® Copolymers as Novel Carriers for Colon-Specific Drug Delivery"

_polymers, 2020, doi:10.3390/polym12071459_

Round 1
Reviewer 1 Report
The present manuscript investigates the "Interpolymer complexes of Eudragit® copolymers as novel carriers for colon-specific drug delivery". The manuscript compromised of some major concern as mentioned below:
- For the preparation of API loaded IPC, surprisingly no information was provided.
- The manuscript lacks vital information of formation of IPC such as duration of mixing, process of isolation of the precipitates, etc.
- What is the rationale for selecting the ethanol, tetrahydrofuran and isopropanol-acetone mixture solvents in the study.
- No information was provided regarding precipitation of IPC, i.e., factors affecting the precipitate formation
- Interestingly no optimization of precipitation was performed or shared which can significantly alter the properties of the precipitate and hence the complete study.
- What is the effect of 2 days of vacuum drying and ball milling/grinding on the physicochemical properties of the precipitate, polymer and API?
- No information on the ball milling parameters was provided.
- Ball milling can alone significantly alter the Tg value of the formulation. However, no information or investigation was performed in order to validate the same.
- No information on the stability study of the developed IPC was reported. It will be interesting to see the effect of stability study in the release profile of the API or more precisely the physicochemical properties of IPC altering the release profile of API.
- No information of the Tg of the IPC loaded with API was provided. Does it impact the overall Tg of the system?
- The information of DSC must be supported by PXRD analysis.
- Authors are requested to clarify, why the Tg of the system was even higher as compared to the individual polymer Tg, which is very strange.
- Author should also evaluate the effect of at least two loadings on the swelling index and release profile of API, in order to verify the change in release profile is mainly driven by polymer erosion.
- Authors compacted the IPC without any disintegrating excipients, which is not in accordance with the realistic approach. The incorporation of the disintegrating agent in the compact will demonstrate the actual swelling of the polymer in different media. Which could have actually hindered due to very limited permeation of the solvent system within the compact.
- What is the porosity of the compacts?
- Authors must perform the contact angle analysis which can demonstrate the wettability of the polymer.
- What is the particle size distribution of the preblend/premix before the development of compacts?
- What is the rationale for selecting the USP III and IV apparatus?
- The quantification of the soluble fraction of the polymer in the media will be interesting to evaluate the exact mechanism of API release.
- Authors are requested to further elaborate on the details of the method of IPC preparation and other investigations wherever required.
Author Response
Response to Reviewer 1 Comments
Point 1: For the preparation of API loaded IPC, surprisingly no information was provided.
For dissolution testing, physical mixture of IPC and API was prepared and then flat-faced compacts were prepared by powder compression of this mixture. The information on the method of compact preparation and parameters of compression is presented in section 2.7.
Point 2: The manuscript lacks vital information of formation of IPC such as duration of mixing, process of isolation of the precipitates, etc.
Some additional information is included in section 2.2.
Point 3: What is the rationale for selecting the ethanol, tetrahydrofuran and isopropanol-acetone mixture solvents in the study
Using of organic solvents can significantly simplify the synthesis of IPC compared to aqueous media, since there is no need to control the pH of the solutionsÑŽ this is now emphasised in section 3.1.
Point 4: No information was provided regarding precipitation of IPC, i.e., factors affecting the precipitate formation
Some additional information is included in section 2.2. There are many factors that could affect the precipitation of IPC, however, in this study we used specific conditions described in section 2.2 to produce solid IPC samples.
Point 5: Interestingly no optimization of precipitation was performed or shared which can significantly alter the properties of the precipitate and hence the complete study.
We did study the effects of order of addition and solvent nature on the preparation and compositions of IPCs. We believe that this optimisation is sufficient for the purpose of this study. There are several other factors that will affect precipitation (e.g. concentration of polymers in solution, presence of small molecules in solution, temperature, etc). The study of all these factors is definitely outside of the scope of this publication and will warrant a separate study.
Point 6: What is the effect of 2 days of vacuum drying and ball milling/grinding on the physicochemical properties of the precipitate, polymer and API?
Vacuum drying aims to remove organic solvent from the samples. Potentially this affects the glass transition temperature of the IPC as small quantities of organic solvents may cause plasticisation and reduction of glass transition temperature. In this study we have conducted drying for 2 days to ensure complete removal of organic solvents.
Point 7: No information on the ball milling parameters was provided
Ball milling parameters are now included in section 2.2
Point 8: Ball milling can alone significantly alter the Tg value of the formulation. However, no information or investigation was performed in order to validate the same.
We agree that ball milling can affect the Tg. However all the samples were ball milled under identical conditions to ensure that these could then be compared. The study of the effects of ball milling parameters on the Tg is outside of the scope of this manuscript.
Point 9: No information on the stability study of the developed IPC was reported. It will be interesting to see the effect of stability study in the release profile of the API or more precisely the physicochemical properties of IPC altering the release profile of API.
We agree that the study of stability will be of interest. We may perform this study in the future and report it separately. A comment about it is now included in the conclusions.
Point 10: No information of the Tg of the IPC loaded with API was provided. Does it impact the overall Tg of the system?
Compacts of IPC with API were prepared by compression of physical mixture IPC and API. Therefore it is not expected to affect the Tg of the polymer because the drug was not molecularly dispersed in the IPC and is not expected to get involved in the specific interactions with the polymers.
Point 11: The information of DSC must be supported by PXRD analysis.
We do not see any rational of using PXRD analysis to study the IPC. Both Eudragits are amorphous polymers and the IPC formed are also expected to be amorphous. PXRD will not be able to provide any valuable information in this case.
Point 12: Authors are requested to clarify, why the Tg of the system was even higher as compared to the individual polymer Tg, which is very strange.
The Tg of IPC was not higher than the Tg of Eudragit S100 (173.8 °C), which was expected. Typically miscible polymeric blends exhibit Tg that is between the Tg values of individual polymers forming this blends. This is exactly what we have observed in this study.
Point 13: Author should also evaluate the effect of at least two loadings on the swelling index and release profile of API, in order to verify the change in release profile is mainly driven by polymer erosion.
Thank you for this suggestion. Potentially it is of interest for further optimisation of dosage forms. However, we believe it is outside of the scope of this manuscript.
Point 14: Authors compacted the IPC without any disintegrating excipients, which is not in accordance with the realistic approach. The incorporation of the disintegrating agent in the compact will demonstrate the actual swelling of the polymer in different media. Which could have actually hindered due to very limited permeation of the solvent system within the compact.
The compacts in this study were formulated with the aim of drug delivery to the colon. Therefore disintegration agents were not added in these formulations as they will result in their premature disintegration in the stomach.
Point 15: What is the porosity of the compacts?
The porosity of the compacts was not studied in the present study. We do not expect anything interesting from evaluation of the porosity in this case
Point 16: Authors must perform the contact angle analysis which can demonstrate the wettability of the polymer.
Potentially the evaluation of the contact angle could give some information on the hydrophobicity/hydrophilicity of the compact surfaces. We expect to see some correlation with the swelling properties. However, these experiments would make sense only in the case where many more different samples are produced and evaluated. With the number of samples produced in this study we do not see any rational of studying them using contact angle measurements.
Point 17: What is the particle size distribution of the preblend/premix before the development of compacts?
We did not study the particle size distribution in our work. Potentially this could warrant a separate study. It is now mentioned in conclusion section
Point 18: What is the rationale for selecting the USP III and IV apparatus?
With USP III and USP IV in vitro dissolution systems we can have a sequential media change that is essential for evaluating the performance of formulations in conditions simulating the GI tract- i.e. the pH and GI fluid composition in each GI compartment.
Point 19: The quantification of the soluble fraction of the polymer in the media will be interesting to evaluate the exact mechanism of API release.
We agree that quantification of the soluble fraction will be of interest to elucidate the mechanism of API release. If these formulations will be pursued further for commercialisation then this could be studied in the future.
Point 20: Authors are requested to further elaborate on the details of the method of IPC preparation and other investigations wherever required.
More details are added on the method of IPC preparation. Additionally we added a statement on potential further studies in the manuscript.

Reviewer 2 Report
The paper deals with ‘“Interpolymer complexes of Eudragit copolymers as novel carriers for colon-specific drug delivery”. This can be an interesting and scientifically relevant to publish in Polymers. However, few alterations could be made.
- Authors stated “The present work was focused on the use of organic solvents (isopropanol: acetone mixture, ethanol and tetrahydrofuran). This approach has led to the IPC, whose compositions and properties are substantially different from the polycomplexes formed from these copolymers in aqueous media” Besides that, what will be advantages of using the organic solvents approach as compared to aqueous media.
- At “Preparation of solid IPCs” part, it is also recommended to Include more details on the mixing time, how many washes with water, ball mill conditions, etc.
- Table 1 presented IPC-4, IPC-5, IPC-6, however, the results did not include the IPCs.
- Experimental design in Table 1, what was the purpose of comparing between IPC-1, and IPC-2. It seems to be designed for the order of mixing as between IPC-3 and IPC-4, however, quite many variations for the ICP-1 and ICP-2.
- Thermal analysis, glass transition temperatures were determined using the reversing heat flow signals. A full scan condition of heat and cool are needed, authors described only heating.
- As authors described in the “Thermal analysis”, all DSC measurements were performed in triplicate. The results are only a value, without SD.
- Lines 177 to 220 can be revised and embedded in the discussion, otherwise this part can be moved to the introduction.
- Lines 258-259, “These values are in good agreement with the values reported in the literature [19]”. Line 268-269, “The value of Tg is increased with increase in the amount of Eudragit® S100 in the IPC structure, and this is in agreement with our previous observations [19, 22]. 269”. Can author provide explanations about the changes in the results rather just a statement “in agreement with the literature”.
- Figure 5, author can provide the opinions, explanation why swelling behavior IPC-1 and IPC-2 is pH independent, and IPC-3 is different.
- Line 361-362, authors stated “Under all conditions tested, release of IND from formulations based 361 on IPC-1 was higher compared to IND release from IPC-2 and IPC-3 formulations”. Explanation the difference?
- Line 389, the authors seem to have over-emphasized the use of their IPCs as carriers for colon-specific drug delivery system.
- Finally, please do a thorough editing effort in the English language and make sure that format, spaces, units and proper grammar rules are followed throughout the manuscript.
Author Response
Response to Reviewer 2 Comments
Point 1: Authors stated “The present work was focused on the use of organic solvents (isopropanol: acetone mixture, ethanol and tetrahydrofuran). This approach has led to the IPC, whose compositions and properties are substantially different from the polycomplexes formed from these copolymers in aqueous media” Besides that, what will be advantages of using the organic solvents approach as compared to aqueous media.
Using of organic solvents can significantly simplify the synthesis of IPC compared to aqueous media, since there is no need to control the pH of the solutions. The additional comments are included in the manuscript.
Point 2: At “Preparation of solid IPCs” part, it is also recommended to include more details on the mixing time, how many washes with water, ball mill conditions, etc.
Additional details are included in section 2.2.
Point 3: Table 1 presented IPC-4, IPC-5, IPC-6, however, the results did not include the IPCs.
Additional results are now included in Figures 1-3
Point 4: Experimental design in Table 1, what was the purpose of comparing between IPC-1, and IPC-2. It seems to be designed for the order of mixing as between IPC-3 and IPC-4, however, quite many variations for the IPC-1 and IPC-2.
Samples IPC-1 and IPC-2 differ in the mixing order and the ratio of the initial copolymers. Since we established that the order of mixing does not have an effect on the composition of the IPC these samples could be compared to evaluate effect of the initial mixture.
Point 5: Thermal analysis, glass transition temperatures were determined using the reversing heat flow signals. A full scan condition of heat and cool are needed, authors described only heating.
Some additional description is included in section 2.6.
Point 6: As authors described in the “Thermal analysis”, all DSC measurements were performed in triplicate. The results are only a value, without SD.
SD are included in the glass transition temperature data given in Table 1. Additionally SD values are now included in Figure 3.
Point 7: Lines 177 to 220 can be revised and embedded in the discussion, otherwise this part can be moved to the introduction.
These lines are revised and additional explanation on the selection of specific organic solvents is included.
Point 8: Lines 258-259, “These values are in good agreement with the values reported in the literature [19]”. Line 268-269, “The value of Tg is increased with increase in the amount of Eudragit® S100 in the IPC structure, and this is in agreement with our previous observations [19, 22]. 269”. Can author provide explanations about the changes in the results rather just a statement “in agreement with the literature”.
This explanation is now provided.
Point 9: Figure 5, author can provide the opinions, explanation why swelling behavior IPC-1 and IPC-2 is pH independent, and IPC-3 is different.
Swelling behavior of IPC-3 depends on the composition of this sample. In the case of IPC-3, the sample contains a larger amount of EPO copolymer, soluble at pH below 5. As indicated in the discussed results, some amount of EPO in the case of this sample is lost (leaching from the compacts) during the experiment, which in turn leads to a different swelling behavior of this IPC. A brief explanation is included in the manuscript.
Point 10: Line 361-362, authors stated “Under all conditions tested, release of IND from formulations based 361 on IPC-1 was higher compared to IND release from IPC-2 and IPC-3 formulations”. Explanation the difference?
Release of IND from formulations based on IPC-1 is higher due to the fact that compacts of IPC-1 with IND are disintegrated during the experiment more than compacts based on IPC-2 and IPC-3, and as we have proved, the driving force for the IND release from investigated samples is a surface erosion of the compact (page 14).
Point 11: Line 389, the authors seem to have over-emphasized the use of their IPCs as carriers for colon-specific drug delivery system.
Thank you for this comment. We have added the word “potentially” claiming that our IPC could be used as carriers for colon-specific drug delivery system. This will help to avoid overemphasizing this claim.
Point 12: Finally, please do a thorough editing effort in the English language and make sure that format, spaces, units and proper grammar rules are followed throughout the manuscript.
The manuscript was carefully proofread and corrected.

Reviewer 3 Report
This article detailed the investigation of using interpolymer complexes (IPCs) of Eudragit® EPO and Eudragit® S100 as a carrier for oral controlled drug delivery to the colon. The experimental methods were well documented and thoughtfully planned out. Extensive material characterization was performed, swelling and release studies were conducted, the results were thoroughly analyzed, and sound conclusions were draw. Overall this manuscript is publishable after the authors correct the following minor concerns:
- All figures and tables are useful and acceptably formatted, although the authors should consider reformatting table 2 to make the units and values line up more uniformly.
- Remove the redundant “for 2 days” from line 86.
- Move Table 1 to the results section, as it is more appropriate for it to be in that section.
- Reorganize the results section so that figures and tables are included in a logical order which align with what is being discussed.
- To further clarify: lines 198 – 225 do a good job of summarizing the results of the study but are immediately followed by sections dedicated to the results from each step of the methods. The authors could reduce redundancy, better concatenate the steps of their work, and reduce the overall length of the manuscript through reformatting the results section.
Author Response
Response to Reviewer 3 Comments
Point 1: All figures and tables are useful and acceptably formatted, although the authors should consider reformatting table 2 to make the units and values line up more uniformly.
Table 2 is reformatted and is moved to Supplementary information
Point 2: Remove the redundant “for 2 days” from line 86.
The redundant “for 2 days” is removed
Point 3: Move Table 1 to the results section, as it is more appropriate for it to be in that section.
Table 1 is moved to Results and discussion.
Point 4: Reorganize the results section so that figures and tables are included in a logical order which align with what is being discussed.
Results were reorganised. Table 1 is moved to Results and Discussion. Tables 3 and 4 were moved to Supplementary Information. Now all figures and tables are in the relevant Results and Discussion sections.
Point 5: To further clarify: lines 198 – 225 do a good job of summarizing the results of the study but are immediately followed by sections dedicated to the results from each step of the methods. The authors could reduce redundancy, better concatenate the steps of their work, and reduce the overall length of the manuscript through reformatting the results section.
The manuscript was reorganised and re-optimised. Some data were moved to Supplementary information. Introductory part in section 3.1 is modified.
